# Incorporation of Net Radiation Model Considering Complex Terrain in Evapotranspiration Determination with Sentinel-2 Data

**Linjiang Wang** [1,2], **Bingfang Wu** [1,2,*], **Abdelrazek Elnashar** [1,2,3], **Weiwei Zhu** [1], **Nana Yan** [1], **Zonghan Ma** [1], **Shirong Liu** [4] **and Xiaodong Niu** [5]

1 State Key Laboratory of Remote Sensing Science, Aerospace Information Research Institute, Chinese Academy of Sciences, Beijing 100094, China; wanglj@radi.ac.cn (L.W.); abdelrazek@aircas.ac.cn (A.E.); zhuww@radi.ac.cn (W.Z.); yannn@radi.ac.cn (N.Y.); mazh@radi.ac.cn (Z.M.)
2 College of Resources and Environment, University of Chinese Academy of Sciences, Beijing 100049, China
3 Department of Natural Resources, Faculty of African Postgraduate Studies, Cairo University, Giza 12613, Egypt
4 Research Institute of Forest Ecology, Environment and Protection, Chinese Academy of Forestry, Beijing 100091, China; liusr@caf.ac.cn
5 Research Institute of Forest Resources Information Techniques, Chinese Academy of Forestry, Beijing 100091, China; 15652573232@163.com
* Correspondence: wubf@radi.ac.cn; Tel.: +86-010-6485-5689

**Abstract:** Evapotranspiration (ET) is the primary mechanism of water transformation between the land surface and atmosphere. Accurate ET estimation given complex terrain conditions is essential to guide water resource management in mountainous areas. This study is based on the ETWatch model driven by Sentinel-2 remote sensing data at a spatial resolution of 10 m incorporating a net radiation model considering the impact of a complex terrain. We tested our model with two years of data in two regions with a high relief near the Huairou (2020) and Baotianman (2019) weather stations. Regarding the validation results of the ET model, the coefficient of determination ($R^2$) reached 0.84 in Huairou and 0.86 in Baotianman, while the root mean square error (RMSE) value reached 0.59 mm in Baotianman and 0.82 mm in Huairou. The validation results indicated that the model is applicable in regions with a complex terrain, and the ET results can capture topographic textures. In terms of the slope aspect, the ET value on south-facing slopes is higher than that on north-facing slopes in both study areas. Accurate ET monitoring in mountainous regions with a high relief yields a profound meaning in obtaining a better understanding of the characteristics of heat and water fluxes at different vegetation growth stages and underlying surface types, which can provide constructive suggestions for water management in mountainous areas.

**Keywords:** evapotranspiration (ET); net radiation; complex terrain; remote sensing



## 1. Introduction

Evapotranspiration (ET), mainly comprising plant transpiration and soil evaporation, is the essential pathway of water transformation between the land surface and atmosphere, linking changes in surface water, carbon cycling, and surface energy [1–3]. Factors such as meteorological, vegetation, and radiation conditions and soil moisture influence the ET process [4,5]. In mountainous areas with a complex topography, topographic factors require greater consideration to obtain better ET estimates [6,7].

Regarding meteorological factors, air pressure and temperature and water vapor pressure tend to decrease with increasing elevation, wind speed, and precipitation [8–10]. In addition, radiation conditions can vary considerably from slope to slope [11,12]. In general, slopes facing south in the Northern Hemisphere are considered sunny slopes, while slopes facing north are considered shady slopes [13]. Sunny slopes tend to receive more solar

radiation than do shady slopes, affecting the regional air temperature distribution and airflow processes [14,15]. The mountain range distribution and orientation can result in differences in temperature between the two sides of mountain ranges, with higher and more variable temperatures on sunny slopes and lower and less variable temperatures on shady slopes due to prevailing radiation conditions [11]. In terms of the wind speed, the situation varies greatly according to the slope location. Wind speeds tend to be higher at summits, ridges, and canyon windrows and lower in basins, valleys, and leeward locations [16,17]. Different slope orientations also exert a notable impact on the precipitation distribution, with mountain tops generally exhibiting a lower temperature and water vapor pressure, with subsequently more clouds and fog, and a higher relative air humidity than do the two sides of mountain ranges, and more precipitation occurs on the windward side than on the leeward side [18,19]. This difference in precipitation can lead to differences in the vegetation landscape between the two sides of mountain ranges, mainly in terms of the vegetation type and biodiversity [20–22]. The distinct radiation, meteorological, and vegetation conditions at different elevations, slopes, and slope orientations result in more complex distributions of surface water and heat fluxes in mountainous regions than those in regions characterized by flat surfaces. Vegetation growth is also a determinant factor to ET, usually monitored with vegetation indices-based methods in literature [23,24]. Moreover, vegetation indices are found to be positively correlated with some hydro-climatic factors like precipitation [25,26]. Mountainous areas account for approximately 24% of the global land area and provide a variety of ecological services, such as water conservation, fertilization, carbon sequestration, oxygen release, atmospheric purification and biodiversity maintenance [27–29]. Consequently, accurate ET monitoring in mountainous areas is crucial and can provide necessary guidance for ecosystem water management in regions with a complex topography [30,31]. The topographic effect can bring the noise to ET retrieval, especially for areas with high relief. According to previous studies, the topographic effect can be reduced by band ratios like NDVI, due to the spectrum similarity between Near InfraRed (NIR) and visible bands [23,24].

Reference ET ($ET_0$) is an important hydrological parameter referred to as the potential ET of well-watered grassland of an assumed relatively uniform height, which is crucial to the estimation of actual ET [32,33]. The $ET_0$ estimation models available in the literature may be broadly classified as (1) fully physically based combination models that account for mass and energy conservation principles [34–36]; (2) semi-physically based models that deal with either mass or energy conservation [37,38]; and (3) black-box models based on artificial neural networks, empirical relationships, and fuzzy and genetic algorithms [39–41]. Nevertheless, the underlying surface for mountainous regions may not be grass, and the environmental factors are not fully ideal. So, it is more practical to focus on the measurement and estimation of actual ET. Among the commonly implemented methods for field-based ET estimation, lysimeters can provide ET measurements at a 1-m resolution, eddy covariance (EC) instruments can provide ET estimates at a 100-m resolution, and large-aperture scintillation (LAS) can provide ET values at a 1-km resolution [42,43]. However, observations obtained with in-situ observation methods are often spatially representative of a limited range and generally only capture underlying surfaces at small scales [44], and in-situ observation instruments are often expensive and exhibit high operating costs. Moreover, it is difficult to effectively characterize regional water fluxes at large scales because the density of observational sites established in mountainous areas with a complex topography is lower than that in areas with a flat terrain due to the harsh environment [45]. Fortunately, the advent of remote sensing technology has facilitated regional ET observations.

Slope-scale ET monitoring at a high spatiotemporal resolution can reflect the water consumption of vegetation on different slope surfaces, and is beneficial to the development and management of water resources in mountainous areas [46,47]. Current ET estimation methods based on remote sensing data mainly include empirical methods [48–50], energy balance residual methods [51–53], and methods based on Penman–Monteith (PM) equations [54–56]. Empirical methods tend to exhibit a simple model structure, but the physical

meaning of the ET process remains ambiguous. Energy balance residual methods are often limited by the low spatial resolution of the thermal infrared band [57]. Methods based on PM equations are mechanistic approaches but only require meteorological data, surface net radiation, and surface resistance to water vapor transmission as input data. With the continuous development of optical remote sensing, high-spatial resolution satellites, such as Sentinel-2, can facilitate ground observations with a spatial resolution of 10 m, which provides a fine picture of underlying surface conditions. Moreover, the revisit period of Sentinel-2 is shorter than that of Landsat series satellites. With a shorter revisit period, the key parameters in ET calculation, such as the normalized difference vegetation index (NDVI) and surface albedo, which exhibit a limited variability during a short period [58,59], can be extended to the daily scale via data interpolation in day-by-day ET simulations.

Accurate estimation of the spatial distribution of net radiation is essential for ET calculation [60,61]. The topography of the underlying mountainous surface is complex, and the solar incidence conditions vary with the slope gradient. Moreover, the surrounding topography imposes a shading effect on the slope surface, resulting in considerable differences in the received incident radiation between various slope gradients and aspect directions. Therefore, it is necessary to consider the influence of terrain factors on the incident solar radiation in net radiation calculations to obtain better ET estimates. The availability of a high-resolution digital elevation model (DEM), e.g., TanDEM-X, which exhibits a spatial resolution of 0.4 arcsec (approximately 12 m), allows the surface relief characteristics of mountainous areas to be reflected in greater detail, thus facilitating the acquisition of high-spatial resolution surface radiation data. To our knowledge, no studies have incorporated a high-resolution DEM in remote sensing-based ET calculations.

Considering the above reasons, this study aims to estimate ET with the ETWatch model at a high spatial resolution (10 m) based on Sentinel-2 and TanDEM-X data to help decisionmakers and land and water managers develop suitable water resource management strategies in areas with a complex topography. This work follows two specific objectives: (1) to develop a daily net radiation-terrain-based model and (2) to estimate ET, considering remote sensing data, in different agroecosystems in areas with a complex terrain. The key findings could be helpful for land and water managers in similar areas.

## 2. Materials and Methods

### 2.1. Study Area

This study was performed in two regions covering the Huairou and Baotianman stations. Huairou is located in the southern part of the Yan Mountains in northeastern Beijing in northern China, which experiences a warm temperate semihumid continental monsoon climate with four distinct seasons, with both rain and heat during the same period. Consequently, warm and humid conditions prevail in summer, while cold and dry conditions occur in winter. The average annual temperature ranges from 9 °C to 13 °C, and the average annual precipitation ranges from 600 to 700 mm, mainly concentrated in the period from June to August. The Huairou station is located in the southeastern part of Huairou, and the longitudinal and latitudinal coordinates of the site are 116°39′35″ E and 40°25′22″ N, respectively, at an altitude of 328 m. Baotianman is located in the eastern part of the Qinling Mountains, on the southern slope of the Funiu Mountains in Neixiang County, Henan Province, Central China, which belongs to the transitional area from the northern subtropical zone to the warm temperate zone, and exhibits the transitional characteristics of the eastern monsoon zone, with a monsoonal continental climate and four distinct seasons. Summer is hot, winter is cold, the temperature quickly rises in spring, the annual average temperature reaches 15.1 °C, and the annual average precipitation reaches 855.6 mm. Baotianman station is located within the Baotianman Nature Reserve, with geographical coordinates of 111°56′07″ E and 33°29′59″ N and an altitude of 1410.7 m. The first selected study area encompassed the Huairou station, covering approximately 400 km², and the second selected study area encompassed the Baotianman station, covering

approximately 225 km$^2$ (Table 1 and Figure 1). We denoted these two study areas, i.e., Huairou and Baotianman, as HR and BTM, respectively.

**Table 1.** Basic features of two study areas.

| Name | Geographical Location | Altitude (m) | Climate |
|---|---|---|---|
| Huairou (HR) | 116°39′35″ E, 40°25′22″ N | 328 | Continental monsoon climate |
| Baotianman (BTM) | 111°56′07″ E, 33°29′59″ N | 1410.7 | |

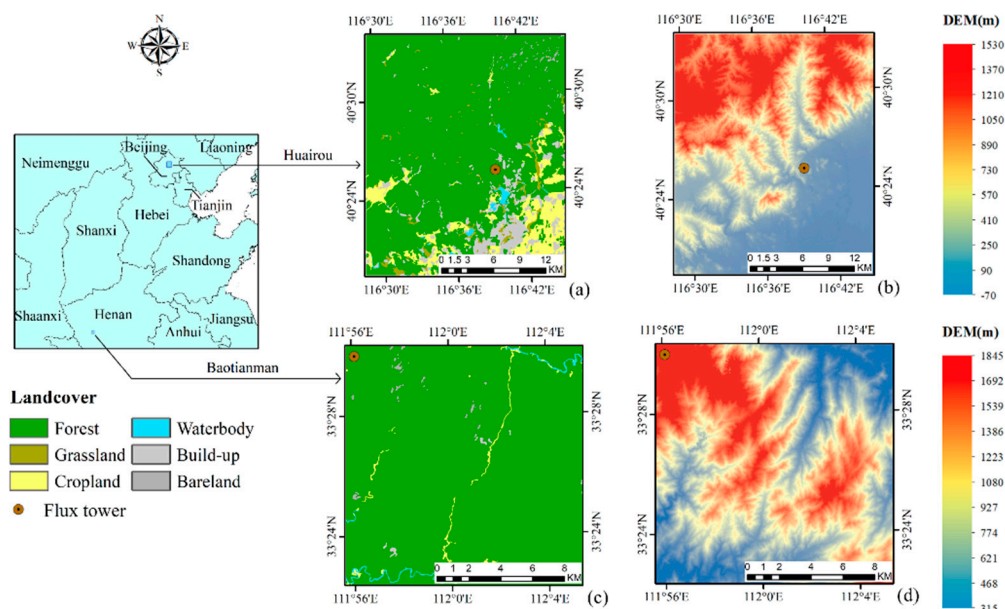

**Figure 1.** Study area (land cover spatial distribution in HR (**a**) and BTM (**c**) and DEM of HR (**b**) and BTM (**d**)).

### 2.2. Data Sources

The data considered in this study mainly include remote sensing data, in-situ EC measurement data, meteorological data, elevation data and certain auxiliary data, like landcover, as described in the following subsections. Table 2 shows the summary of the dataset used in this study.

**Table 2.** Summary of dataset used in this study.

| Dataset | Resolution | | Source |
|---|---|---|---|
| | Temporal | Spatial | |
| Sentinel-2 | 5-day | 10 m | https://scihub.copernicus.eu/dhus/#/home (accessed on 2 December 2021) |
| FY-2F | hourly | 1.25 km (VIS) 5 km (NIR) | http://satellite.nsmc.org.cn/PortalSite/Default.aspx (accessed on 5 December 2021) |
| MOD11A1 | daily | 1 km | https://ladsweb.modaps.eosdis.nasa.gov/ (accessed on 5 December 2021) |
| AIRS | daily | 13.5 km | https://disc.gsfc.nasa.gov/ (accessed on 7 December 2021) |
| NCEP | daily | 2.5° | https://psl.noaa.gov/data/gridded/data.ncep.reanalysis2.html (accessed on 5 December 2021) |
| Enhanced SMAP | 3-day | 10 km | https://gimms.gsfc.nasa.gov/SMOS/SMAP/ (accessed on 7 December 2021) |
| In-situ EC data | half hourly | - | field observatory and ChinaFlux http://www.chinaflux.org/ (accessed on 10 December 2021) |
| In-situ meteorological data | half hourly | - | https://data.cma.cn/ (accessed on 28 November 2021) |
| Elevation | - | 0.4 arcsec | https://tandemx-science.dlr.de/ (accessed on 15 November 2021) |
| Landcover | - | 30 m | AIRCAS |

### 2.2.1. Remote Sensing Data

The remote sensing data utilized in this study largely include Sentinel-2, Moderate Resolution Imaging Spectroradiometer (MODIS), Fengyun-2F (FY-2F), Atmospheric Infrared Sounder (AIRS), National Centers for Environmental Prediction (NCEP) and Soil Moisture Active Passive (SMAP) data. Sentinel-2 comprises two multispectral imaging satellites, 2A and 2B, launched successively, carrying Multispectral Imager (MSI) sensors with an orbital altitude of 786 km and an amplitude of 290 km. These satellite sensors cover 13 spectral bands with a ground resolution from 10 to 60 m in each data band, ranging from visible and near-infrared to shortwave infrared (SWIR), with different spatial resolutions. The revisit period is 10 days for one satellite and 5 days for two satellites, considered together. In this study, Sentinel-2 1C-level data were downloaded from the European Space Agency (ESA) Copernicus Data Centre website (https://scihub.copernicus.eu/dhus/#/home (accessed on 2 December 2021)). Atmospheric, radiometric, and geometric corrections of the downloaded 1C-level data were performed with the Sen2Cor plug-in in Sentinel Application Platform (SNAP) software with terrain correction and BRDF correction options. The images were mosaicked and clipped to obtain the surface reflectance in the two study areas. Surface reflectance data in the near-infrared and red bands were obtained to calculate the NDVI, which was considered to simulate the vegetation cover and leaf area index (LAI) and subsequently applied in canopy conductance and soil evaporation calculations. We chose Sentinel-2 rather than Satellite products such as AVHRR, MODIS and Himawari because these satellite products cannot meet the needs of ET estimation in regions with complex terrain due to relatively low spatial resolution. As for Landsat, the temporal resolution is not satisfactory, which cannot provide enough monitoring during the main growing season. The surface albedo was calculated through multiband fitting and applied in subsequent net radiation calculations. Combined with a cloud mask, the Savitzky–Golay (S-G) filtering method was chosen to temporally extend the NDVI and albedo on cloud-free days to daily scales, which has been widely applied in relevant studies [62,63]. Cloud classification data based on FY-2F geostationary satellite data acquired from the China National Satellite Meteorological Center (http://satellite.nsmc.org.cn/PortalSite/Default.aspx (accessed on 5 December 2021)) were employed to estimate sunshine hours [64]. MODIS land surface temperature data products (MOD11A1) at a 1-km spatial resolution were collected from the Level-1 and Atmosphere Archive and Distribution System Distributed Active Archive Center (LAADS DAAC) (https://ladsweb.modaps.eosdis.nasa.gov/ (accessed on 5 December 2021)). Observations provided by the AIRS installed on the Aqua satellite of the Goddard Earth Sciences Data and Information Services Center (GES DISC) (https://disc.gsfc.nasa.gov/ (accessed on 7 December 2021)) provided vertical distribution data of the air temperature and humidity, and the height of the atmospheric boundary layer was determined according to a previous study [65]. The air temperature and humidity at the height of the atmospheric boundary layer were retrieved from AIRS data, while the wind velocity was determined with NCEP reanalysis data obtained from the Physical Sciences Laboratory of the National Oceanic and Atmospheric Administration (NOAA) (https://psl.noaa.gov/data/gridded/data.ncep.reanalysis2.html (accessed on 5 December 2021)) [66]. National Aeronautics and Space Administration (NASA)–US Department of Agriculture (USDA) enhanced SMAP global soil moisture data provided global information at a 10-km spatial resolution and were mainly employed for daily surface conductance reconstruction [67].

### 2.2.2. In-Situ Tower Observation Data

The in-situ observations obtained at flux towers mainly included radiation component observation and EC data. The radiation component observation data were considered to validate the net radiation-terrain-based model, while the EC data were employed to validate the resulting ET model estimates. In this study, we collected data in HR in 2020 and BTM in 2019. Observation data pertaining to HR were collected from the field observatory, while BTM data were provided by ChinaFLUX (http://www.chinaflux.org/ (accessed on

10 December 2021)). The adopted EC observation instruments largely comprised ultrasonic anemometers and $CO_2/H_2O$ infrared gas analyzers. The observation tower in HR is 40 m high, while the observation tower in BTM is 36 m high. The EC instruments were set up at a height of 30 m in HR and 29 m in BTM, while the radiation component observation instruments were placed at 20 m in HR and 22 m in BTM. The data sampling frequency of the EC observations at the two sites was 10 Hz, which were averaged and stored for 30 min. The underlying surface surrounding the HR observation tower is dominated by arborvitae, while oak trees dominate the underlying surface surrounding the BTM observation tower. Since high-resolution ET values are calculated in this study, it was necessary to average the model calculation results based on the footprint area of each EC instrument to match the flux observation results during model validation. This study applied a flux footprint prediction (FFP) model to calculate the footprint area [68], and the results are shown in Figure 2. The collected EC data were processed following the literature [44], including the exclusion of data outliers, data before and after precipitation, and nighttime data under extremely low-turbulence conditions (the friction velocity is lower than 0.2 m/s). It should be noted that approximately 88 days of EC data from May to August 2020 were missing in HR because of instrument or data logger issues.

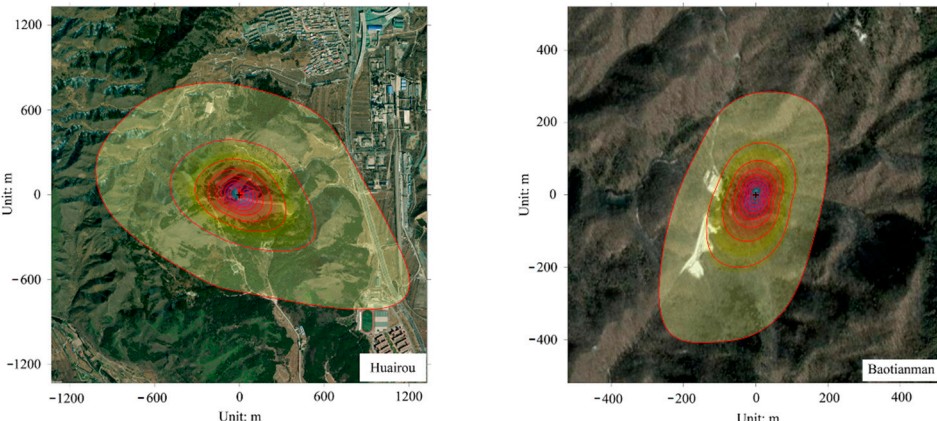

**Figure 2.** Footprint area of the two EC instruments.

2.2.3. Meteorological Data

The first part of the meteorological data encompassed daily meteorological data obtained from the China Meteorological Data Service Center (https://data.cma.cn/ (accessed on 28 November 2021)). The data at each station mainly include the relative air humidity (RH), wind speed ($V_{wind}$), atmospheric pressure (PRS), maximum air temperature ($T_{max}$), minimum air temperature ($T_{min}$), mean air temperature ($T_{mean}$), and sunshine hours (sunt). These parameters were extrapolated from the point scale to the spatial scale with the kriging interpolation method [69]. Sea-level values of four parameters, namely, $T_{max}$, $T_{min}$, $T_{mean}$, and PRS, were first calculated based on the site elevation using empirical relationships between these parameters and elevation [51,70]. After interpolation, the extrapolated sea-level values in all pixels were transformed into values at the corresponding elevation with DEM data. All the interpolated meteorological data were processed to follow the same spatial reference system and pixel size as the remote sensing data. The second part of the meteorological data originated from China Meteorological Radiation Data International Exchange Stations, also obtained from the China Meteorological Data Service Center (https://data.cma.cn/ (accessed on 28 November 2021)). These stations provided daily values of radiation observations, mainly including the total radiation, direct beam radiation, reflected radiation, and diffuse radiation. Observation data were collected at two stations (nearest to HR and BTM) from 2000 to 2020 to calibrate the solar radiation model in the net radiation calculation process.

#### 2.2.4. Elevation Data

The elevation data, namely, TanDEM-X global DEM data, was mainly applied in the net radiation calculation process and provided by the German Aerospace Center (Deutsches Zentrum für Luft- und Raumfahrt or DLR) (https://tandemx-science.dlr.de/ (accessed on 15 November 2021)) with a spatial resolution of 0.4 arcsec (~12 m) [71].

#### 2.2.5. Auxiliary Data

Auxiliary data, including land cover, was obtained in this study. ChinaCover, a land cover dataset for China, was employed in this study and was mostly involved in the analysis of ET model results and parameterization processes such as canopy conductance, which was provided by the Aerospace Information Research Institute, Chinese Academy of Sciences (AIRCAS). ChinaCover contains 38 secondary classes and 6 primary classes (forestland, grassland, cropland, water bodies, built-up land, and bare land) with a spatial resolution of 30 m [72].

#### 2.3. Methods

Figure 3 shows a summary of the input data in this study and provides the proposed slope-scale ET model framework.

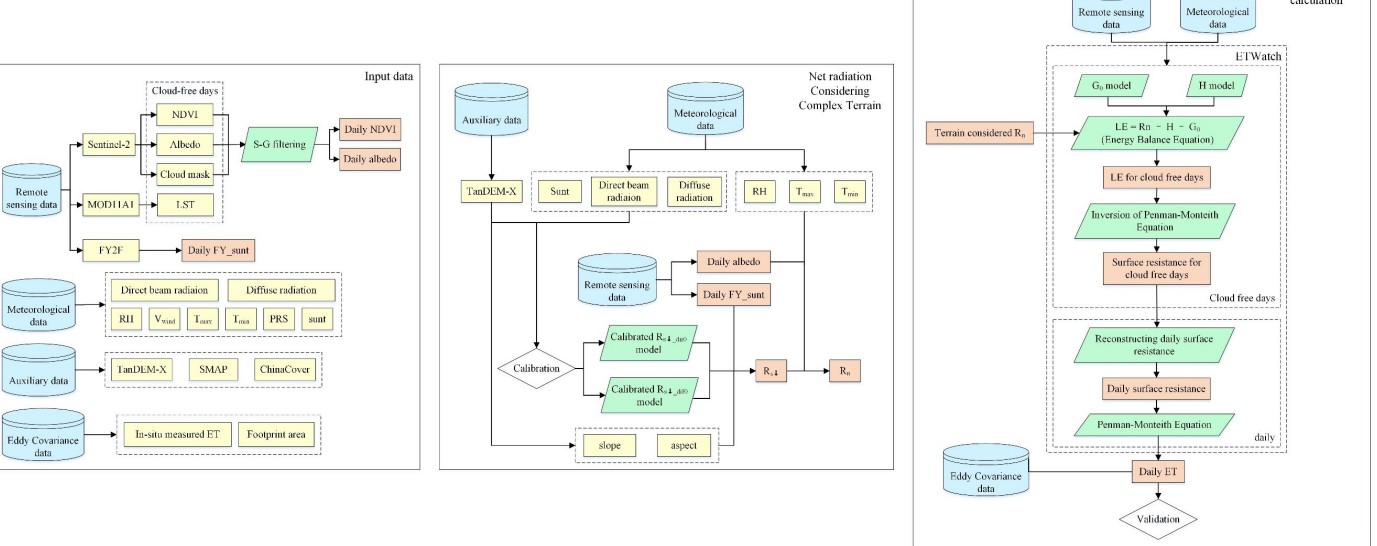

**Figure 3.** Summary of the main input data and model flow chart. The blue disk storage symbols indicate the four types of input data, the yellow rectangles indicate specific input data, the orange rectangles indicate intermediate and output data, and the green parallelograms indicate models and equations.

#### 2.3.1. Net Radiation Calculation

In this study, the net radiation ($R_n$) was calculated with Equation (1) [73].

$$R_n = R_{s\downarrow}(1 - \alpha) - R_{nl} \tag{1}$$

where $R_{s\downarrow}$ is the downward shortwave radiation, $\alpha$ is the surface albedo, and $R_{nl}$ is the net longwave radiation. The downward shortwave radiation ($R_{s\downarrow}$, incident solar radiation) was calculated as the sum of the direct solar radiation ($R_{s\downarrow\_dir}$), sky diffuse radiation ($R_{s\downarrow\_dif}$) and reflected radiation in adjacent regions ($R_{s\downarrow\_adj}$). Hence, the total incident solar radiation $R_{s\downarrow}$ at the surface, considering a complex terrain, can be expressed as Equation (2).

$$R_{s\downarrow} = R_{s\downarrow\_dir} + R_{s\downarrow\_dif} + R_{s\downarrow\_adj} \tag{2}$$

The direct solar radiation ($R_{s\downarrow\_dir}$) was calculated with Equation (3a–e) [74,75].

$$R_{s\downarrow\_dir} = R_{s\downarrow\_dir0} * R_b \tag{3a}$$

$$\alpha_{day} = \sin^{-1}\left(0.85 + 0.3 * \varphi * \sin\left(\frac{2\pi}{365} * DOY - 1.39\right) - 0.42 * \varphi^2\right) \tag{3b}$$

$$\theta_z = \frac{\pi}{2} - \alpha_{day} \tag{3c}$$

$$\phi_s = \cos^{-1}\left(\frac{\sin\alpha_{day} * \sin\varphi - \sin\delta}{\cos\alpha_{day} * \cos\varphi}\right) \tag{3d}$$

$$R_b = \frac{\cos\theta_z * \cos\beta + \sin\theta_z * \sin\beta * \cos(\phi_s - A)}{\cos\theta_z} \tag{3e}$$

where $R_{s\downarrow\_dir0}$ (W/m²) is the direct solar radiation on a horizontal surface, $R_b$ denotes the ratio of the direct solar radiation on an inclined surface to that on a horizontal surface, $\alpha_{day}$ (radians) is the daily average solar elevation angle calculated based on the latitude ($\varphi$, radians) and day of the year (DOY) [75], $\theta_z$ (radians) is the daily solar zenith angle complementary to $\alpha_{day}$, $\phi_s$ (radians) is the daily solar azimuth angle, $\delta$ is the declination angle (radians) and can be calculated with the DOY [73], $\beta$ (radians) is the slope, and A (radians) is the slope orientation. $R_{s\downarrow\_dir0}$ can be estimated with the daily sunshine hours $sunt$ and daily maximum possible sunshine hours $sunt_{max}$ with validated linear, quadratic, or cubic empirical regression models of $R_{s\downarrow\_dir0}/R_a$ and $sunt/sunt_{max}$, expressed as Equation (4a–c) [76,77].

$$\frac{R_{s\downarrow\_dir0}}{R_a} = a_1 + b_1 \times \frac{sunt}{sunt_{max}} \tag{4a}$$

$$\frac{R_{s\downarrow\_dir0}}{R_a} = a_2 + b_2 \times \frac{sunt}{sunt_{max}} + c_2 \times \left(\frac{sunt}{sunt_{max}}\right)^2 \tag{4b}$$

$$\frac{R_{s\downarrow\_dir0}}{R_a} = a_3 + b_3 \times \frac{sunt}{sunt_{max}} + c_3 \times \left(\frac{sunt}{sunt_{max}}\right)^2 + d_3 \times \left(\frac{sunt}{sunt_{max}}\right)^3 \tag{4c}$$

where $R_a$ is the daily extraterrestrial solar radiation (MJ·m⁻²·day⁻¹) calculated according to [73], and the units can be converted into W/m² through multiplication by $\frac{10^6}{24\times3600}$. Moreover, $sunt_{max}$ can be determined based on $\delta$ and $\varphi$, and lowercase letter with numeric subscripts such as $a_1$, $b_2$, and $c_3$ are empirical regression coefficients.

The sky diffuse radiation ($R_{s\downarrow\_dif}$) can be calculated by employing the sky view factor ($\Phi_{sky}$) to correct the diffuse sky radiation on a horizontal surface ($R_{s\downarrow\_dif0}$), expressed as Equation (5a,b) [78,79].

$$R_{s\downarrow\_dif} = R_{s\downarrow\_dif0} \times \Phi_{sky} \tag{5a}$$

$$\Phi_{sky} = V_{sky}/\frac{\pi}{2} \tag{5b}$$

where $V_{sky}$ is the sky view angle (radians). In a single pixel, the minimum sky view angle in eight directions (namely, N, NE, N, NW, W, SW, S, and SE) was calculated, and the average minimum sky view angle was regarded as the sky view angle in this pixel [80]. In each image pixel ($P_0$), the elevation angle from every pixel P within radius of L (L here was set to 10 [81]) along the above eight directions to pixel $P_0$ ($\theta$) was calculated with Equation (6a,b).

$$\theta = \tan^{-1}\left(\frac{H - H_0}{D}\right) \tag{6a}$$

$$D = S_p\sqrt{(X - X_0)^2 + (Y - Y_0)^2} \tag{6b}$$

where $X$ and $Y$ are the image coordinates of pixel P, $X_0$ and $Y_0$ are the image coordinates of pixel $P_0$, $H$ and $H_0$ are the altitudes (m) of pixel P and $P_0$, respectively, and $S_p$ is the pixel size (m). Thus, the corresponding zenith angle ($\theta_0$) can be expressed as Equation (7).

$$\theta_0 = \begin{cases} \frac{\pi}{2} - \theta, 0 \leq \theta < \frac{\pi}{2} \\ \frac{\pi}{2}, -\frac{\pi}{2} \leq \theta < 0 \end{cases} \tag{7}$$

The average minimum zenith angle in eight directions is defined as the sky view angle $V_{sky}$. $R_{s\downarrow\_dif0}$ can also be estimated in a similar manner to the daily sunshine hours *sunt* and daily maximum possible sunshine hours $sunt_{max}$ with validated linear, quadratic or cubic empirical regression models [76,77].

The reflected radiation in adjacent regions ($R_{s\downarrow\_adj}$) was calculated with Equation (8a,b) and the approximation method of Dozier and Frew (1990) [82].

$$R_{s\downarrow\_adj} = F_{ts}\alpha_m \left( R_{s\downarrow\_dir} + R_{s\downarrow\_dif} \right) \tag{8a}$$

$$F_{ts} = \frac{1 + \cos\beta}{2} - \Phi_{sky} \tag{8b}$$

The net longwave radiation ($R_{nl}$) was calculated mainly based on the meteorological parameters, as expressed in Equation (9) [73].

$$R_{nl} = \sigma \frac{T_{min}^4 + T_{max}^4}{2} (0.34 - 0.14\sqrt{e_a}) \left( 1.35\frac{R_{s\downarrow}}{R_{s\downarrow0}} - 0.35 \right) \tag{9}$$

where $\sigma$ is the Stefan-Boltzmann constant ($4.903 \times 10^{-9}$ MJ·K$^{-4}$·m$^{-2}$·day$^{-1}$), $e_a$ is the actual vapor pressure (KPa) and can be calculated with the relative humidity and air temperature [83], $T_{max}$ and $T_{min}$ are the daily maximum and minimum air temperatures (K), respectively, and $R_{s\downarrow0}$ is the clear-sky solar radiation (MJ·m$^{-2}$·day$^{-1}$) and can be calculated with $R_a$ and altitude [73].

The surface shortwave broadband albedo ($\alpha$), determining the incident solar radiation amount absorbed by the underlying surface, was calculated with the linear combination of the Sentinel-2 surface reflectance data using the method reported by Li et al. (2018) [84], thus realizing narrow-to-broadband conversion via Equation (10).

$$\alpha = \sum_{i=0}^{n} a_i x_i + c \tag{10}$$

where $i$ is the band number of the Sentinel-2 images, $x_i$ is the surface reflectance in band $i$, and $a_i$ and $c$ are empirical regression coefficients, as listed in Table 3.

**Table 3.** Surface shortwave broadband albedo conversion coefficients from narrow bands for Sentinel-2 [84].

| Bands | Coefficients |
|---|---|
| Band 2 (Blue) | 0.2688 |
| Band 3 (Green) | 0.0362 |
| Band 4 (Red) | 0.1501 |
| Band 8A (Red Edge 4) | 0.3045 |
| Band 11 (SWIR 1) | 0.1644 |
| Band 12 (SWIR 2) | 0.0356 |
| Constant | −0.0049 |

### 2.3.2. Evapotranspiration Calculation

In this study, the net radiation model considering the influence of terrain, as introduced in Section 2.3.1, was incorporated into the ETWatch model to estimate the ET value at the slope scale in the two study areas.

It is difficult to perform ET calculations on cloudy days. The ETWatch model mainly captures key information on sunny days and combines multisource remote sensing and meteorological data to calculate energy balance components. First, based on the concept of the residual method, the instantaneous energy balance components (net radiation [85], soil heat flux [86], and sensible heat flux [87]) were calculated to determine the instantaneous ET. Second, following the assumption that the evaporation fraction remains constant during the day, the instantaneous ET value on sunny days was extended to the daily ET value on a certain time scale [88]. Then, the surface conductance on sunny days was obtained with an inverted PM equation. Next, the surface conductance on sunny days was extended to the daily scale, combined with meteorological, biophysical, soil and radiation factors [89]. Finally, the daily ET was calculated through the PM equation. To conclude, the ETWatch model integrates a series of submodels, namely, FY sunshine hour (FYsunt) model [90], atmospheric boundary layer (ABL) model [65], aerodynamic roughness length ($A_{z0m}$) model [91], vapor pressure deficit (vpd) model, net radiation ($R_n$) model [85], sensible heat flux (H) model [87], soil heat flux (G) model [86], and gap-filling model of the surface conductance ($r_s$) [89], which jointly realize the fine description of the temporal and spatial distribution patterns of the key parameters in the ET estimation process, thus facilitating improvement of the regional ET estimation reliability [69]. Figure 4 shows the ETWatch model architecture.

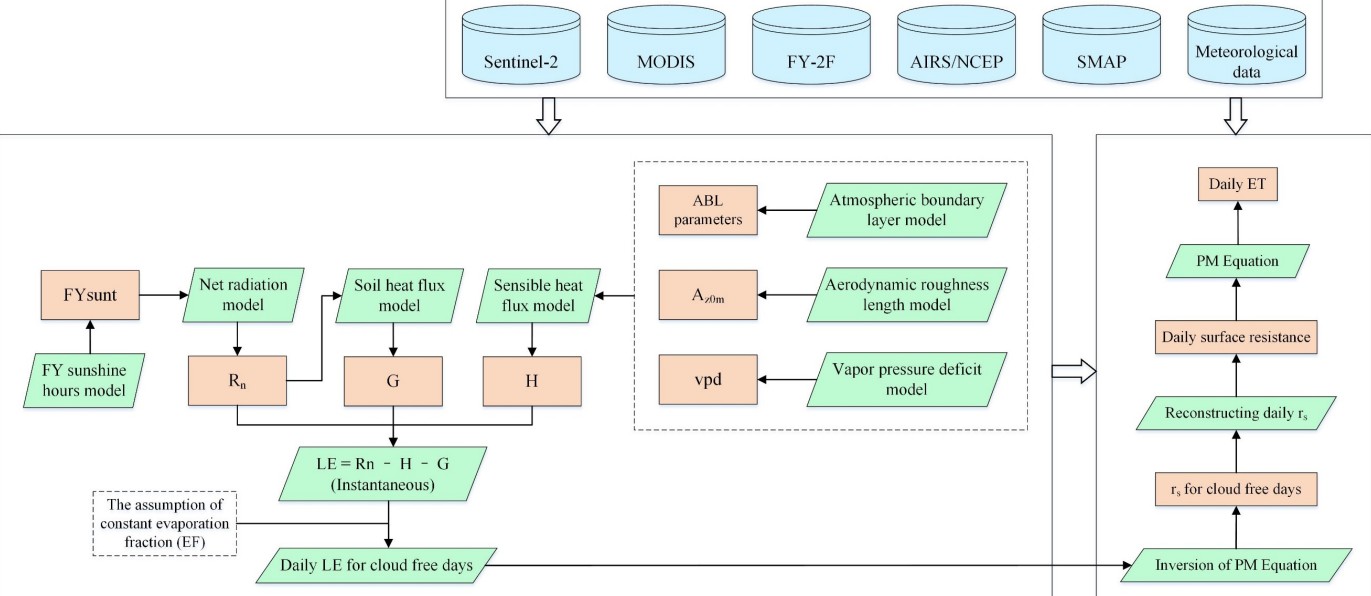

**Figure 4.** Detailed model architecture of ETWatch. The blue disk storage symbols indicate input data, the orange rectangles indicate intermediate and output data, and the green parallelograms indicate models and equations.

## 3. Results

### 3.1. Net Radiation Results

In the net radiation calculation process, this study focused on the influence of terrain factors on solar radiation and divided solar radiation into three parts: direct solar radiation, sky diffuse radiation, and reflected radiation in adjacent regions. Figure 5 shows the spatial distribution of the slope and sky view factor in the two study areas. The sky view factor is also an important indicator reflecting the characteristics of the terrain, and the value

ranges from 0 to 1. A sky view factor value close to 1 suggests a good view of the sky in the surrounding hemisphere, which generally occurs in plain areas, ridges, or mountain peaks, while a sky view factor value close to 0 often occurs in low-lying or valley bottom areas. Concerning the situation surrounding HR, the southeast side of the study area is a plain area, and the main land cover types include farmlands and built-up surfaces. When the terrain is relatively flat, the sky view factor is high, while the sky view factor is low under complex mountainous terrain conditions. In terms of the situation surrounding BTM, most of the study area is located in mountainous areas, and only some narrow topographically flat areas exhibit high sky view factor values.

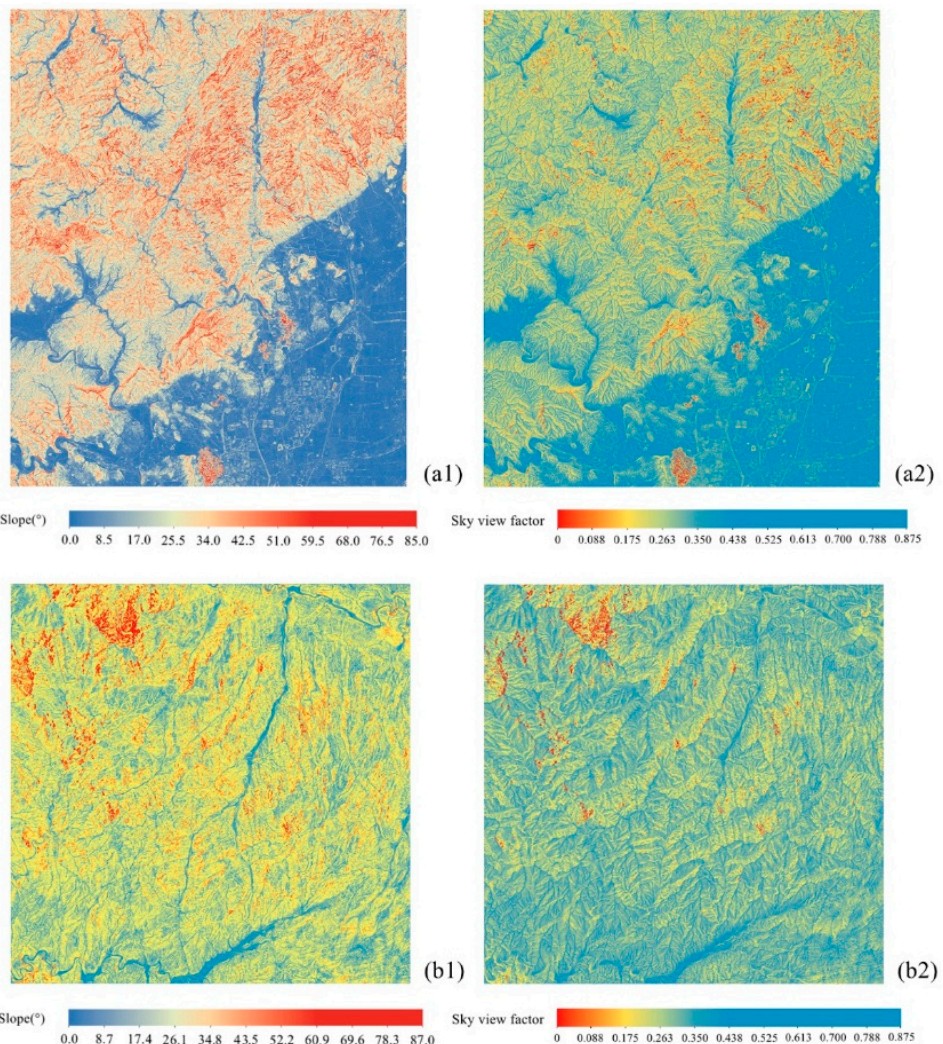

**Figure 5.** Spatial distribution of the slope (**a1**,**b1**) and sky view factor (**a2**,**b2**) in HR (**a1**,**a2**) and BTM (**b1**,**b2**).

Figure 6 shows scatter plots of the sky view factor versus the elevation and slope in the two study areas. The elevation in HR is clearly concentrated within the range from 50–1100 m, and the sky view factor ranges from 0.7–0.875, with the highest sky view factor values in a large plain area at a low elevation (lower than 250 m), as shown in Figure 6a1. The elevation in BTM largely ranges from 450–1300 m, and the sky view factor is mainly concentrated within the range from 0.7–0.8 (please refer to Figure 6b1). As shown in Figure 6a2,b2, the sky view factor in both study areas tends to decrease with increasing slope.

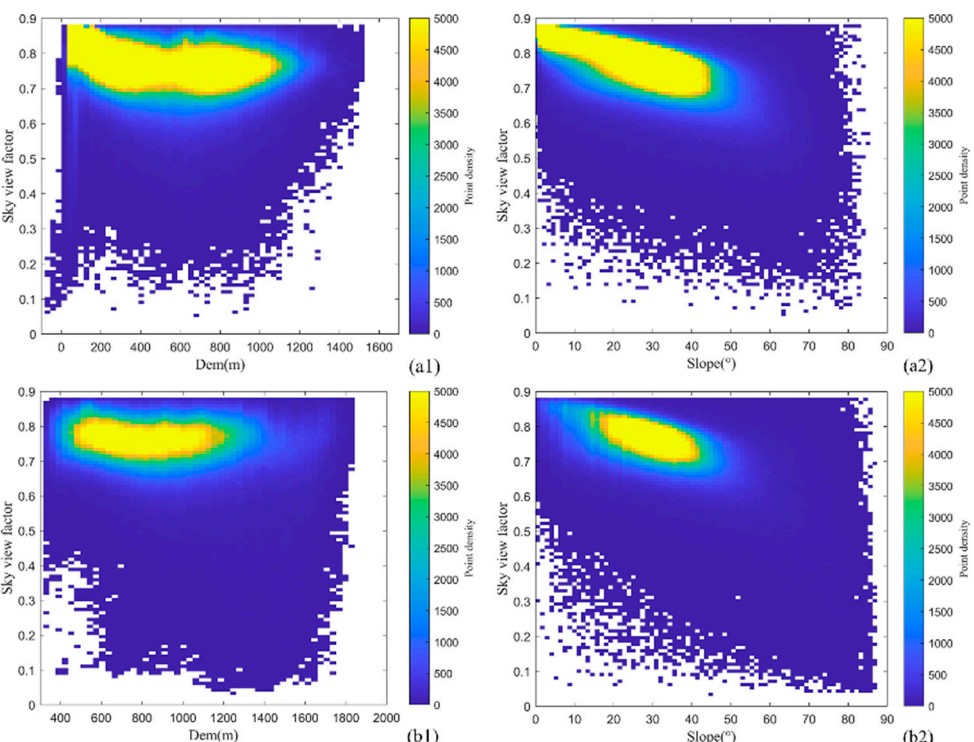

**Figure 6.** Scatter plots of the sky view factor with the DEM (**a1**,**b1**) and slope (**a2**,**b2**) for HR (**a1**,**a2**) and BTM (**b1**,**b2**).

Calibration results of the relationship between $R_{s\downarrow\_dir0}/R_a$ and $R_{s\downarrow\_dif0}/R_a$ and $sunt/sunt_{max}$ are shown in Figure 7, based on radiation observations at the stations nearest the two study areas, and the equations and related statistics are summarized in Table 4. In regard to the horizontal direct radiation ($R_{s\downarrow\_dir0}$), the linear fit was the worst, the quadratic and cubic polynomials were very similar, and the cubic polynomial yielded the best fit. The adjusted coefficient of determination ($R^2$) values were 0.760 and 0.708 for HR and BTM, respectively, while the RMSE values were 0.115 and 0.111, respectively. Regarding the horizontal sky diffuse radiation ($R_{s\downarrow\_dif0}$), the best fit was still obtained with the cubic polynomial, with adjusted $R^2$ values of 0.588 and 0.350 for HR and BTM, respectively, and RMSE values of 0.052 and 0.048, respectively.

The abovementioned calibrated equations with the cubic polynomial were substituted into Equations (3) and (6) to obtain solar and net radiation results, respectively. In the validation process of the net radiation calculation model, the model calculation results were compared to the obtained ground observations (Figure 8), and the accuracy of the net radiation model calculation results in both HR and BTM was satisfactory, with $R^2$ values of 0.87 and 0.85, respectively, and RMSE values of 15.64 and 25.90 W/m$^2$, respectively. With reference to the mean error (ME), the HR results were relatively close to the overall average value (4.45 W/m$^2$), while the BTM results were overestimated (10.54 W/m$^2$). The validation results indicated that the accuracy of the net radiation model calculation results is satisfactory, and the validated model can be adopted in the subsequent ET calculations.

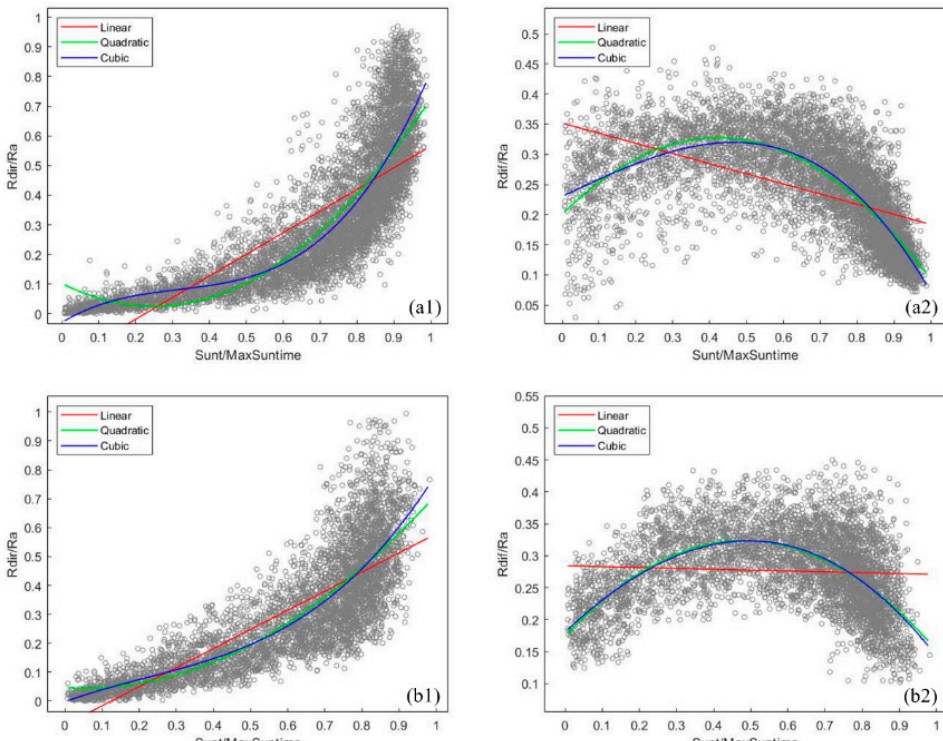

**Figure 7.** Direct (**a1,b1**) and diffuse (**a2,b2**) radiation model calibration results for HR (**a1,a2**) and BTM (**b1,b2**). The red, green, and blue lines indicate linear, quadratic, and cubic regression functions, respectively.

**Table 4.** Summary of the calibration results of the direct and diffuse radiation models for HR and BTM.

| Region | Radiation Type | Regression Type | Equation | Adjusted $R^2$ | RMSE |
|---|---|---|---|---|---|
| Huairou | $R_{s\downarrow\_dir0}$ | Linear | $y = -0.164 + 0.729x$ | 0.636 | 0.141 |
| | | Quadratic | $y = 0.102 - 0.615x + 1.238x^2$ | 0.744 | 0.118 |
| | | Cubic | $y = -0.029 + 0.751x - 1.898x^2 + 1.992x^3$ | 0.760 | 0.115 |
| | $R_{s\downarrow\_dif0}$ | Linear | $y = 0.352 - 0.168x$ | 0.285 | 0.068 |
| | | Quadratic | $y = 0.201 + 0.598x - 0.706x^2$ | 0.581 | 0.052 |
| | | Cubic | $y = 0.230 + 0.287x + 0.008x^2 - 0.454x^3$ | 0.588 | 0.052 |
| Baotianman | $R_{s\downarrow\_dir0}$ | Linear | $y = -0.082 + 0.661x$ | 0.660 | 0.121 |
| | | Quadratic | $y = 0.047 - 0.076x + 0.744x^2$ | 0.700 | 0.113 |
| | | Cubic | $y = -0.002 + 0.469x - 0.610x^2 + 0.930x^3$ | 0.708 | 0.111 |
| | $R_{s\downarrow\_dif0}$ | Linear | $y = 0.284 - 0.014x$ | 0.003 | 0.061 |
| | | Quadratic | $y = 0.175 + 0.617x - 0.639x^2$ | 0.344 | 0.049 |
| | | Cubic | $y = 0.181 + 0.545x - 0.458x^2 - 0.125x^3$ | 0.350 | 0.048 |

Figure 9 shows the day-by-day variation in the calculation results of the net radiation model compared to the in-situ observations, including the downward shortwave radiation. The variation patterns of the net radiation model calculation results were consistent with the in-situ observations in both study areas. As an essential component of the net radiation balance at the surface, the downward shortwave radiation is a major energy source for processes such as plant growth and atmospheric circulation. Figure 9 clearly shows that the downward shortwave radiation greatly influences the net radiation.

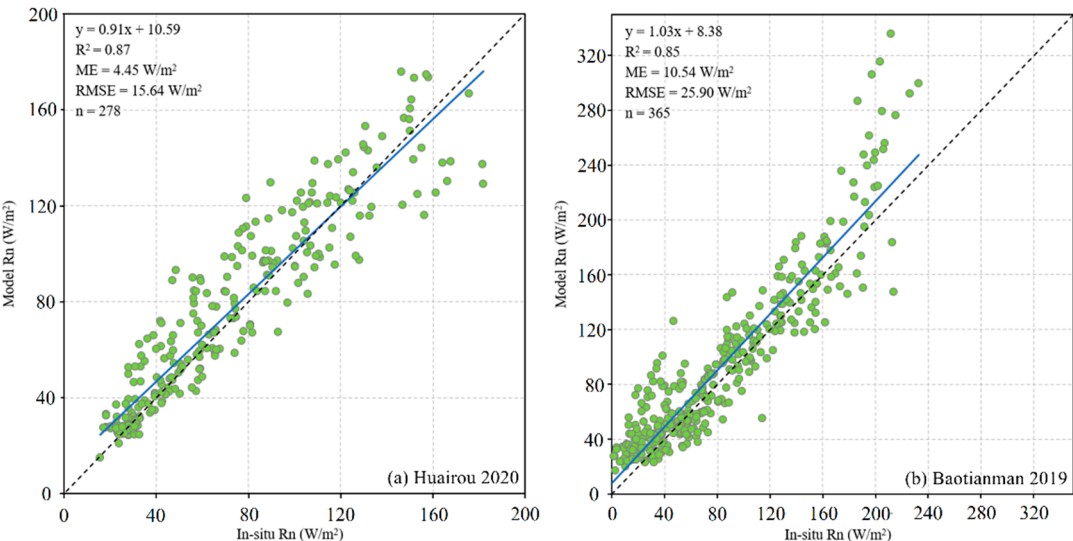

**Figure 8.** Scatter plots of the modeled $R_n$ values with the in-situ $R_n$ values for HR (**a**) and BTM (**b**), combined with a summary of the validation results in the upper left corner, fitted line (blue line), and 1:1 line (black dashed line).

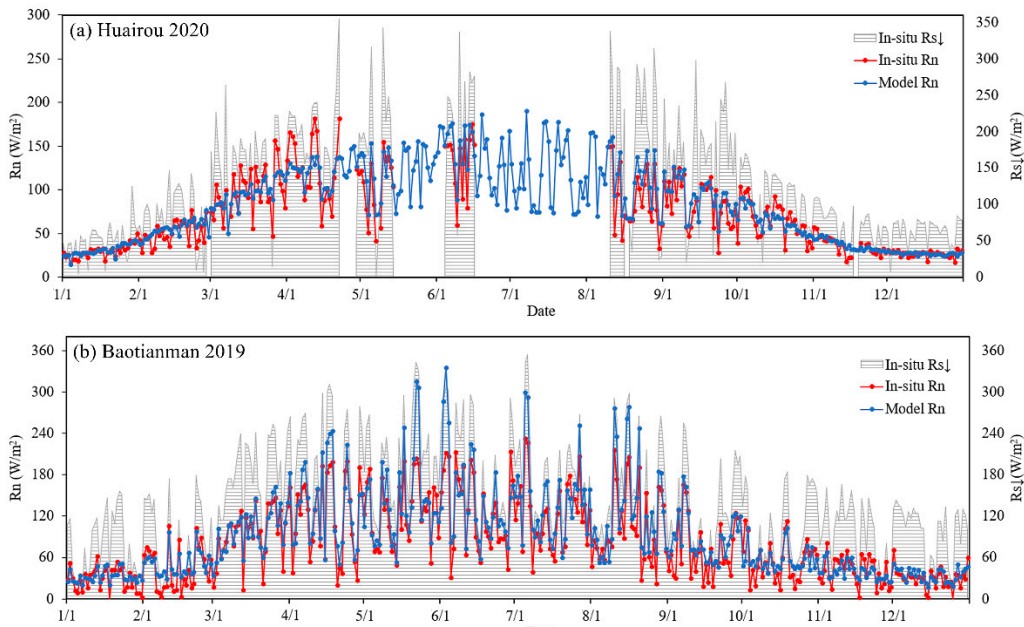

**Figure 9.** Time series of the modeled $R_n$ (blue dotted line) and in-situ $R_n$ (red dotted line) values for HR (**a**) and BTM (**b**), with the in-situ $R_{s\downarrow}$ observations (area with the horizontal gray lines).

Figure 10 shows a comparison of the net radiation model performance in the different months, characterized by the coefficient of determination ($R^2$), relative mean error (rME), and relative root mean square error (rRMSE), between the net radiation model calculation results and in-situ observations. As shown in Figure 10a, the net radiation models in both study areas performed better in summer and autumn than in winter and spring, with the highest correlation coefficient value of 0.9 in August in HR, and exceeding 0.7 in May, June, and September. The highest correlation coefficient value in BTM was 0.85 in April, exceeding 0.7 in all months except January, October, November and December. The net radiation results for HR were overestimated in all months except for March, April, and October, while the net radiation results for BTM were slightly underestimated in November. In terms of rME (Figure 10b), the deviation in BTM was large. Overall, the deviation in

February, July, August, September and December exceeded 15%. The largest deviation in HR occurred in August at 21%, while the absolute values of rME in March, April, October, November and December were less than 10%. As shown in Figure 10c, the variation in rRMSE in HR ranged from 14% (December) to 25% (February), and the variation in rRMSE in BTM ranged from 17% (March) to 44% (February).

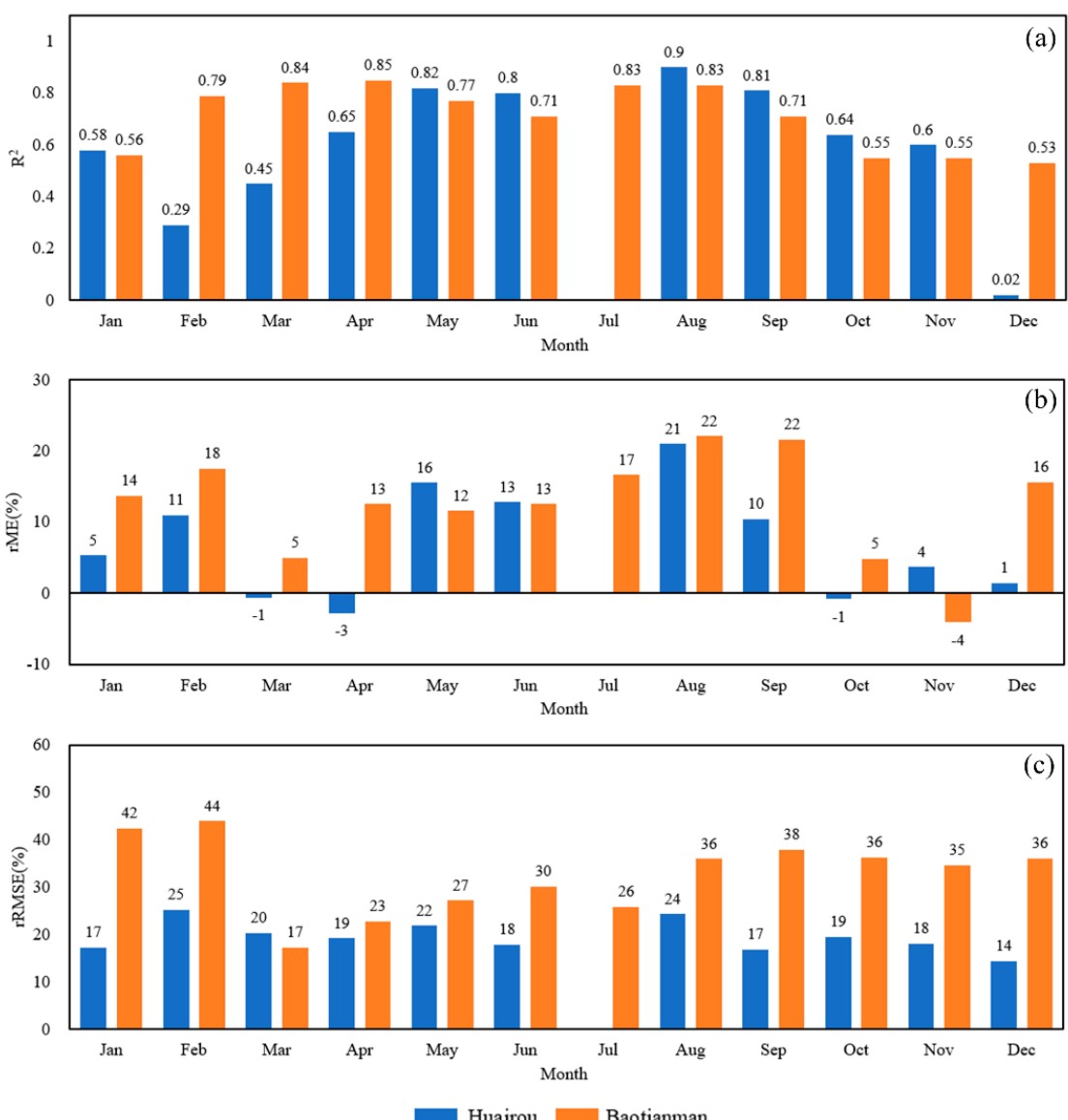

**Figure 10.** Monthly performance of the Rn model expressed by the coefficient of determination, $R^2$ (**a**), relative mean error, rME (**b**), and relative root mean square error, rRMSE (**c**) in HR (blue bar) and BTM (orange bar).

Based on the spatial distribution of the annual average net radiation (Figure 11), the net radiation on built-up surfaces in the plain area to the southeast of HR was low, and the net radiation over water bodies was the highest. Moreover, BTM is mainly mountainous, and the variation in net radiation in this area was low. In both study areas, the variation in net radiation in the vegetation-covered mountainous area could indicate certain textures with the topography, reflecting the influence of the topography on the net radiation to a certain extent.

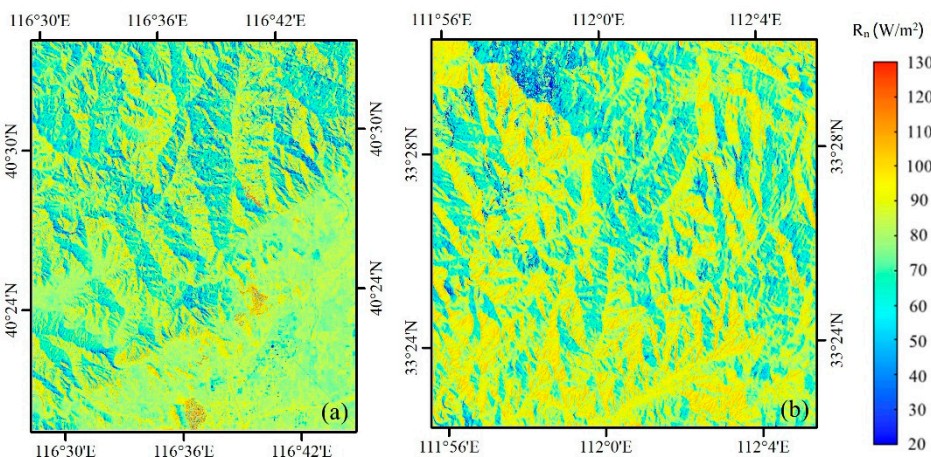

**Figure 11.** Spatial distribution of the yearly average value of the model Rn results in HR in 2020 (**a**) and BTM in 2019 (**b**).

### 3.2. ET Results

Validation of the ET model was accomplished by comparing the model calculation results to EC data, and the validation results are shown in Figure 12a,b. The $R^2$ value was 0.84 for HR and 0.86 for BTM, and from the perspective of deviation, the model calculation results were slightly higher in both study areas, with ME values of −0.39 and 0.11 mm for HR and BTM, respectively. The RMSE was lower in BTM (0.59 mm) than that in HR (0.82 mm). The ET model performed relatively well in both topographically complex study areas.

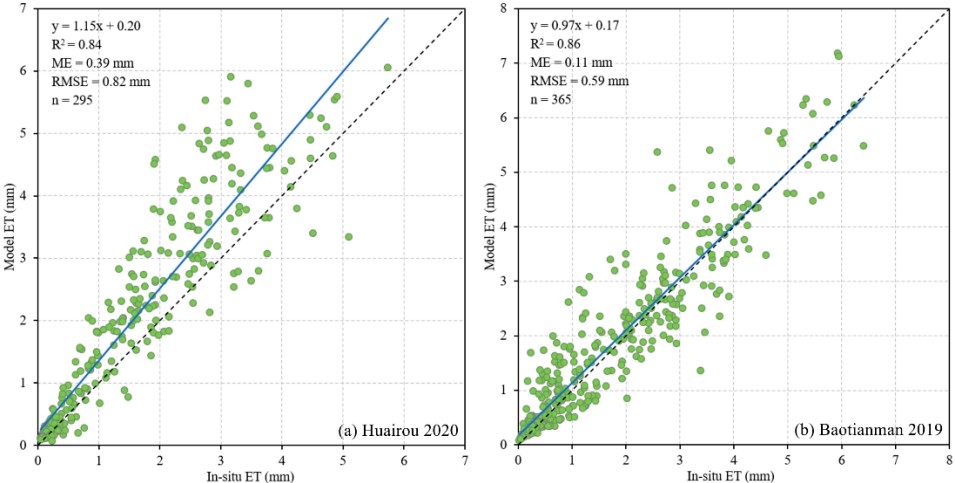

**Figure 12.** Scatter plots of the remote sensing-based ET values with the in-situ ET data for HR (**a**) and BTM (**b**), combined with a summary of the validation results in the upper left corner, fitted line (blue line), and 1:1 line (black dashed line).

Figure 13 compares the day-by-day variation in the ET model results to the in-situ observations, including the net radiation. Again, the day-by-day variation in the ET model results for both study areas suitably agreed with that in the in-situ observations. The ET variation pattern was similar to that of the net radiation. Net radiation is an essential component of the surface energy balance, and as the energy source of the latent heat flux (ET), this parameter highly influences ET.

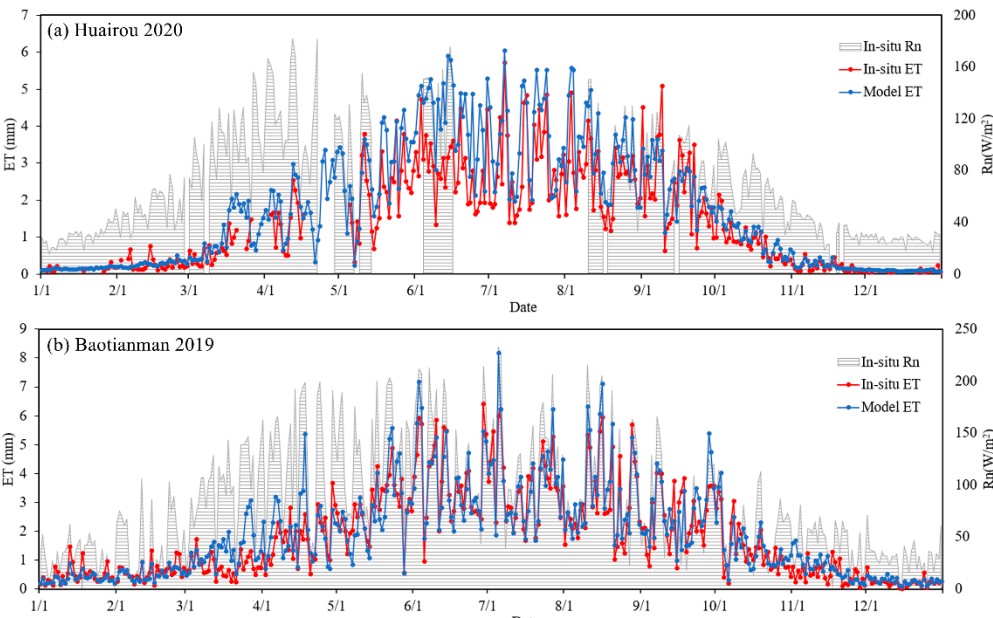

**Figure 13.** Time series of the remote sensing-based ET values (blue dotted line) and in-situ ET (red dotted line) data for HR (**a**) and BTM (**b**), with the in-situ $R_n$ observations (area with the horizontal gray lines).

Figure 14 shows a comparison of the ET model performance in the different months between the ET calculation results and in-situ observations. As shown in Figure 14a, both study areas exhibited a good performance. The highest $R^2$ value in HR was 0.84 in April, while only January, February, March, November and December yielded $R^2$ values lower than 0.5. The $R^2$ value in August was the highest, namely 0.87 in BTM, and there were only four months with $R^2$ values below 0.4 (March, April, November and December). As shown in Figure 13b, the ET model results were overestimated in February, March, April, and June in HR, while in BTM, the ET model results were underestimated in January, February, May, June, and July. As shown in Figure 14b, the largest deviation (rME) occurred in March (exceeding 70%), while the absolute values from May to October were less than 10%. The model results were all overestimated in HR. The largest deviation (rME) in HR occurred in June (51%), while the absolute values remained within 10% in January, February and September. Regarding rRMSE (Figure 14c), the seasonal variation characteristics were similar to those of rME. The rRMSE values varied between 33% (September) and 82% (March) in HR and between 15% (June) and 102% (March) in BTM. Overall, the accuracy of the ET model results was satisfactory during the main vegetation growing season.

Figures 15 and 16 show month-by-month spatial distributions of the ET model calculation results in 2020 in HR and 2019 in BTM. The monthly ET changes in the two study areas reflected a pronounced seasonality, gradually increasing with vegetation growth, peaking in summer, and gradually decreasing thereafter. Since the net radiation calculation considered the topography influence, the ET results could also reflect the differences between the various topographic conditions to a certain extent.

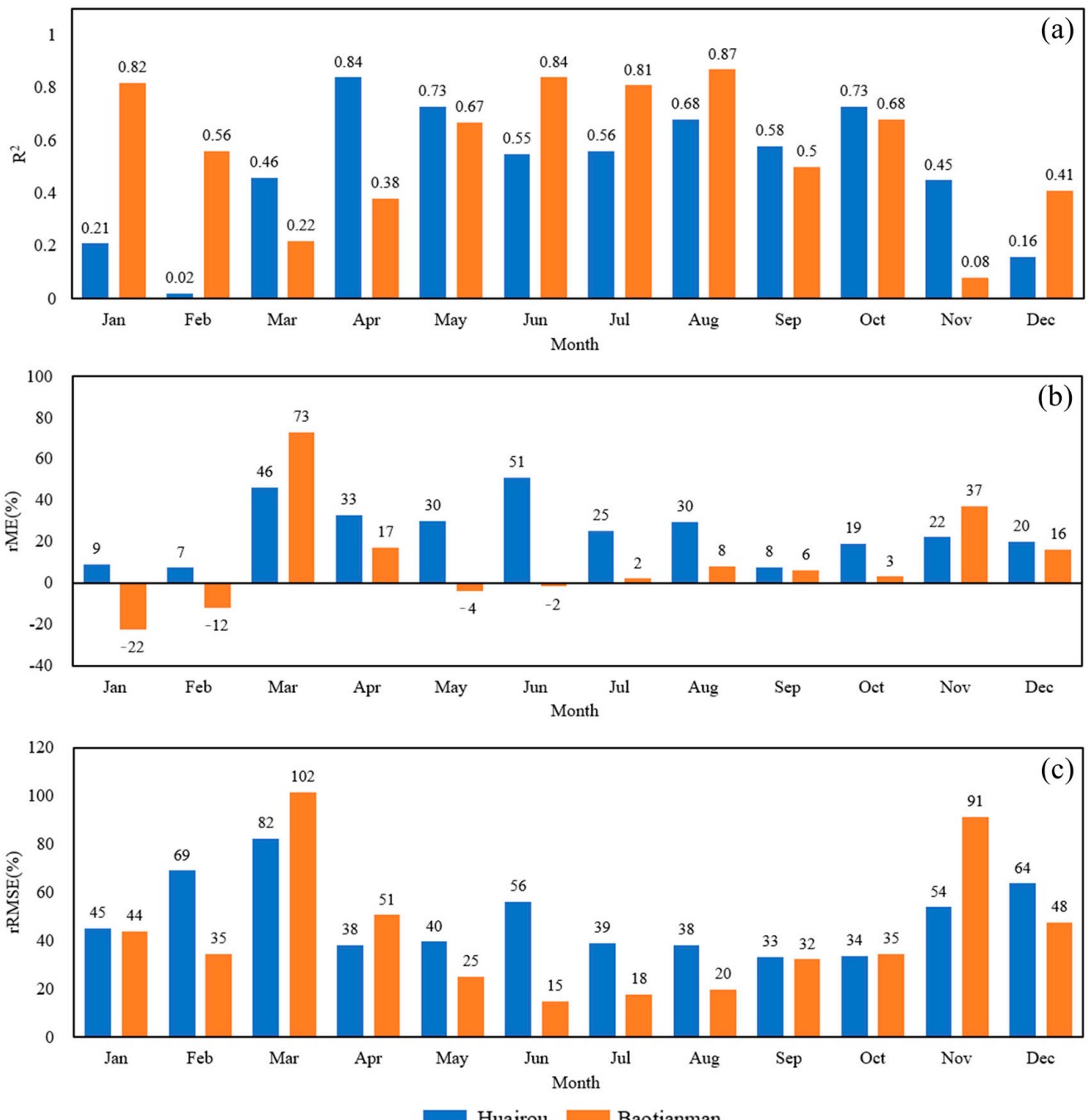

**Figure 14.** Monthly performance of the ET model expressed by the coefficient of determination, $R^2$ (**a**), relative mean error, rME (**b**), and relative root mean square error, rRMSE (**c**) in HR (blue bar) and BTM (orange bar).

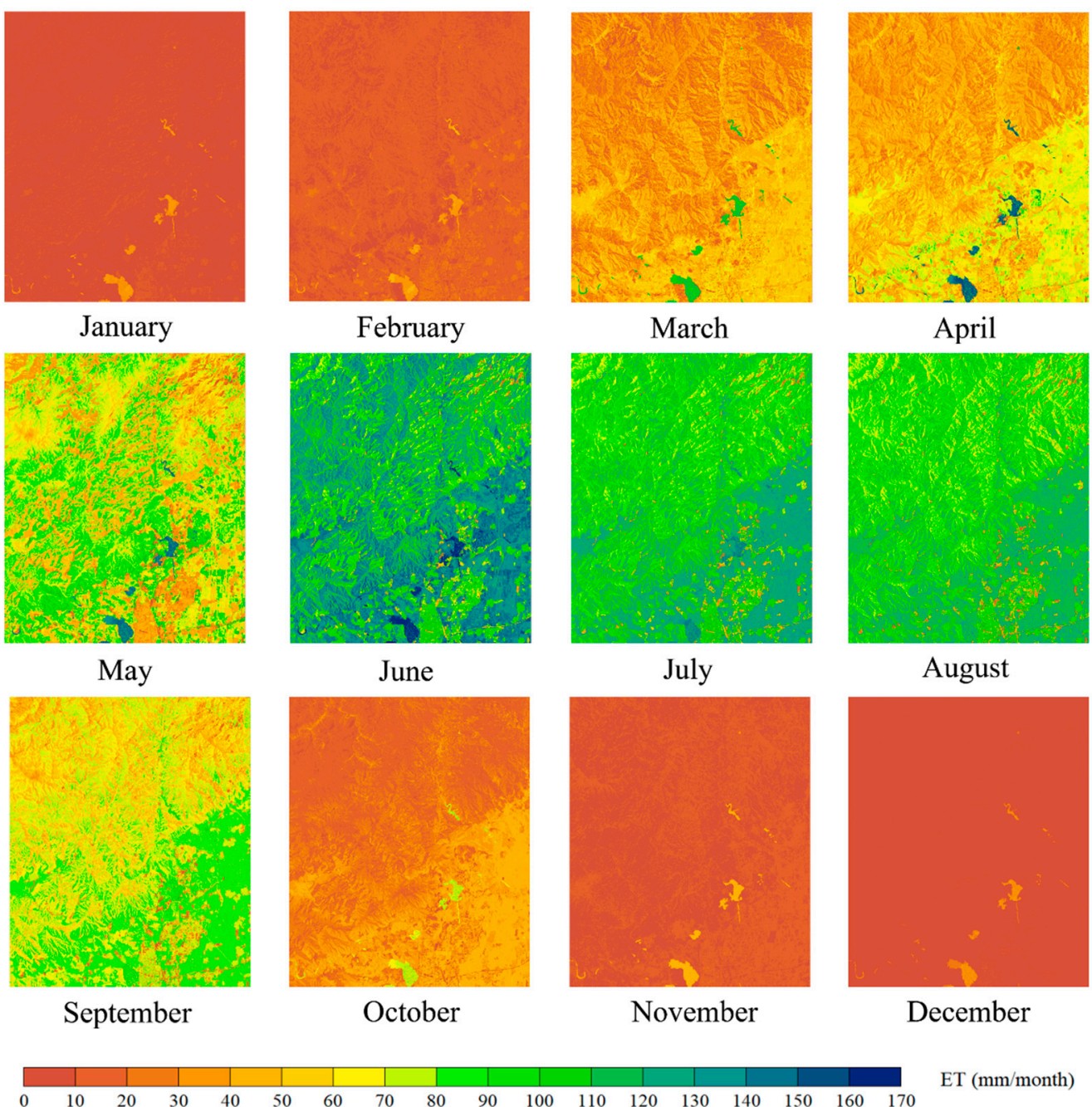

**Figure 15.** Spatial distribution of the monthly ET for HR in 2020.

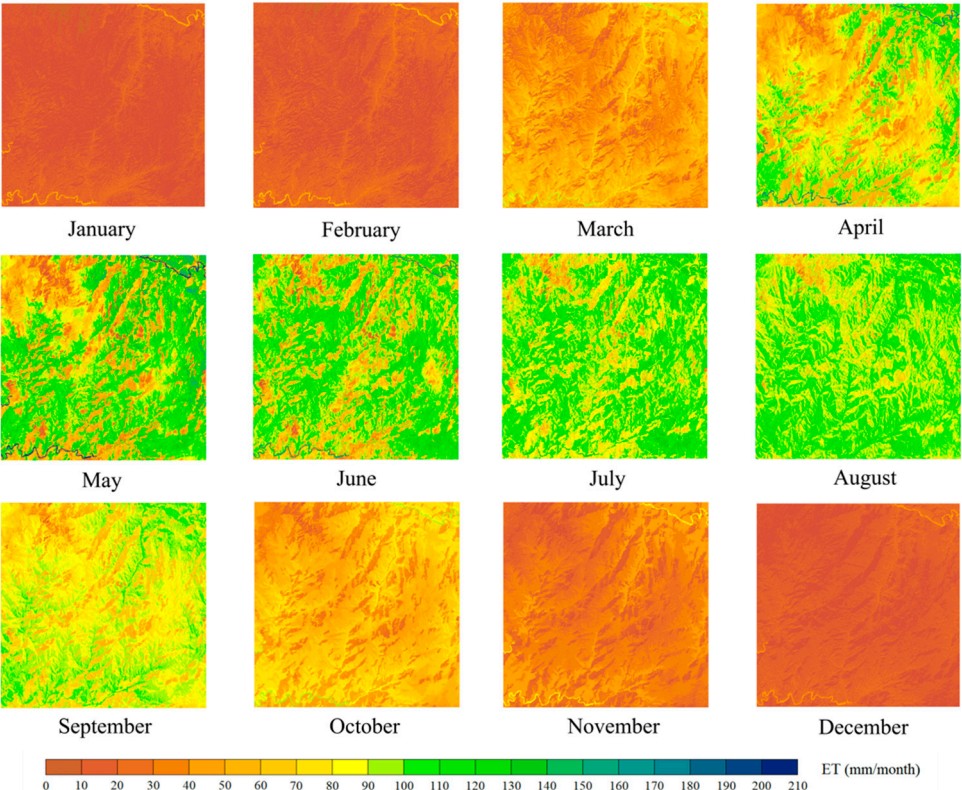

**Figure 16.** Spatial distribution of the monthly ET for BTM in 2019.

## 4. Discussion

### 4.1. Model Performance

In this study, topographic factors, such as the elevation and sky view factor, were introduced to calculate the downward shortwave radiation, and a net radiation model considering topographic factors was constructed. Then, the daily net radiation calculation results were incorporated into a remote sensing based ETWatch model with a high spatial resolution under complex topographic conditions. The model performance was satisfactory, and a similar accuracy was achieved to that reported in previous studies [6,92–94].

The incorporation of high-resolution DEM data, combined with high-resolution Sentinel 2 remote sensing data, provided net radiation results with a high spatial resolution and accuracy. To explore whether the involvement of topographic factors yields more accurate net radiation simulation results, we also calculated the net radiation with the ETWatch model (Model_ETW), which does not account for topographic factors. Comparing the net radiation calculation results of the present model to those of Model_ETW and those based on the Global Land Data Assimilation System (GLDAS) [95], Climate Forecast System version 2 (CFSv2) [96], ERA5 [97], and Clouds and the Earth's Radiant Energy System (CERES) [98] datasets considering the in-situ $R_n$ observations, both study areas obtained the highest accuracy with the model proposed in this study in regard to the correlation coefficient (R) and RMSE (Figure 17). Consideration of topographic factors yielded $R_n$ model results with a higher accuracy than that obtained with Model_ETW, and all $R_n$ datasets in both HR and BTM. The net radiation data mentioned above were incorporated into the ET calculation process, and the ET calculation results were compared in a similar manner, as shown in Figure 18. The results revealed that the ET calculation results based on the $R_n$ model data suitably agreed with the observation data in both study areas, thus generating the best performance. Furthermore, the accuracy of the ET model results was slightly improved with the help of the terrain-considered $R_n$ results.

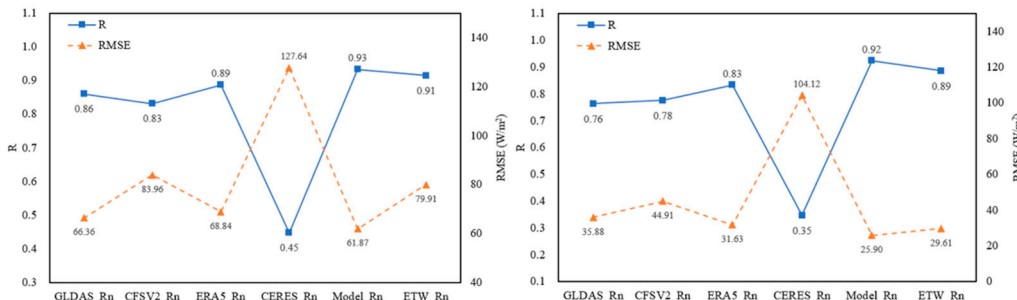

**Figure 17.** Comparison of the accuracy of the $R_n$ values retrieved from the different datasets and the modeled $R_n$ values in this study with the in-situ $R_n$ observation data.

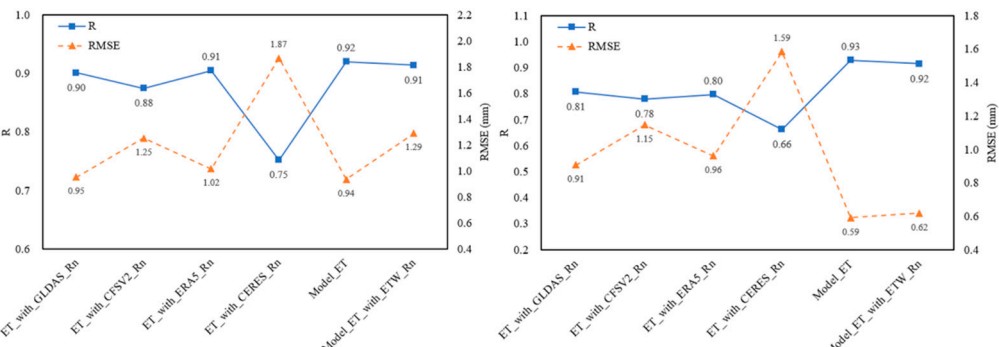

**Figure 18.** Comparison of the accuracy of the ET results with the different Rn datasets as input and the modeled ET values in this study with the in-situ ET observation data.

As shown in Figures 11, 15 and 16, the $R_n$ results and final ET results in this study revealed certain topographic characteristics in terms of the spatial distribution. $R_n$ in both HR and BTM decreased with increasing slope (Figure 19(a1,b1), respectively). The mean net radiation value in HR was 91.84 $W/m^2$ on sunny slopes, 89.90 $W/m^2$ on semisunny slopes, 81.89 $W/m^2$ on semishady slopes and 79.63 $W/m^2$ on shady slopes, while the mean net radiation value in BTM was 91.96 $W/m^2$ on sunny slopes, 89.03 $W/m^2$ on semisunny slopes, 80.75 $W/m^2$ on semishady slopes, and 79.71 $W/m^2$ on shady slopes (Figure 19(a2,b2), respectively). In both study areas, the mean net radiation values were higher on both sunny and semisunny slopes than on shady and semishady slopes, respectively. $R_n$, as the main source of energy in the ET process, also incorporated these topographic features into the final ET results. Figure 20 shows that ET in both study areas also decreased with increasing slope. The mean annual ET value on sunny and semisunny slopes was 616.93 mm, and the mean annual ET value on shady and semishady slopes was 596.22 mm in HR. In contrast, the mean annual ET value on sunny and semisunny slopes reached 761.58 mm, and the mean annual ET value on shady and semishady slopes reached 655.53 mm in BTM. ET on south-facing slopes was higher than that on north-facing slopes in both study areas.

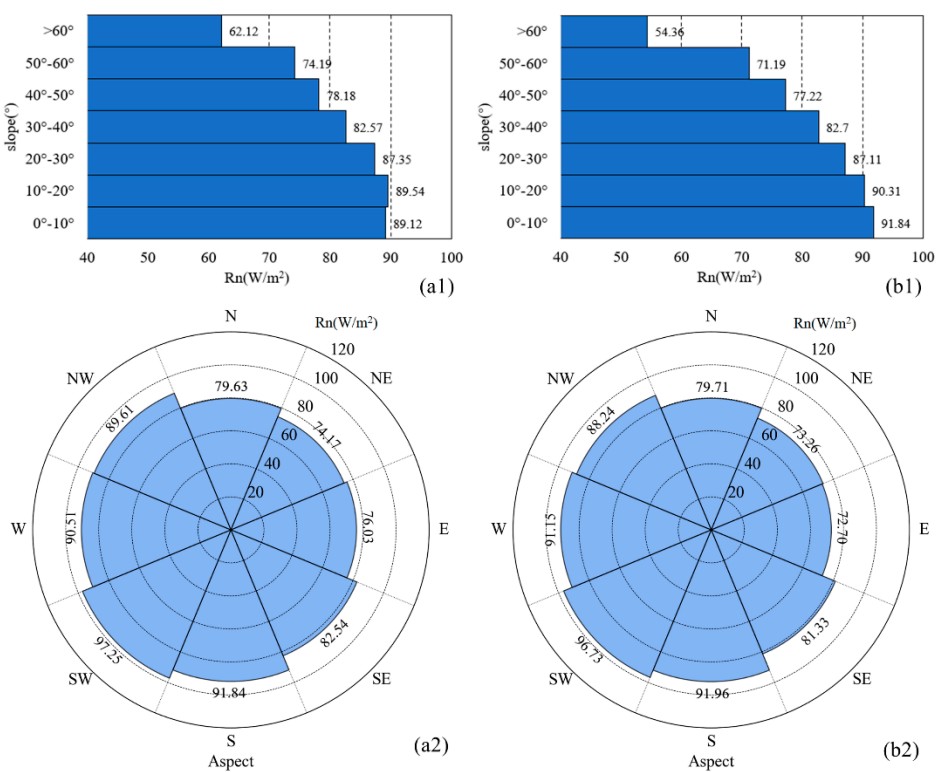

**Figure 19.** Average daily R$_n$ at the different slopes (**a1**,**b1**) and aspect ranges (**a2**,**b2**) in HR (**a1**,**a2**) and BTM (**b1**,**b2**). Aspect range abbreviations such as N, NW and W are explained in Table 5.

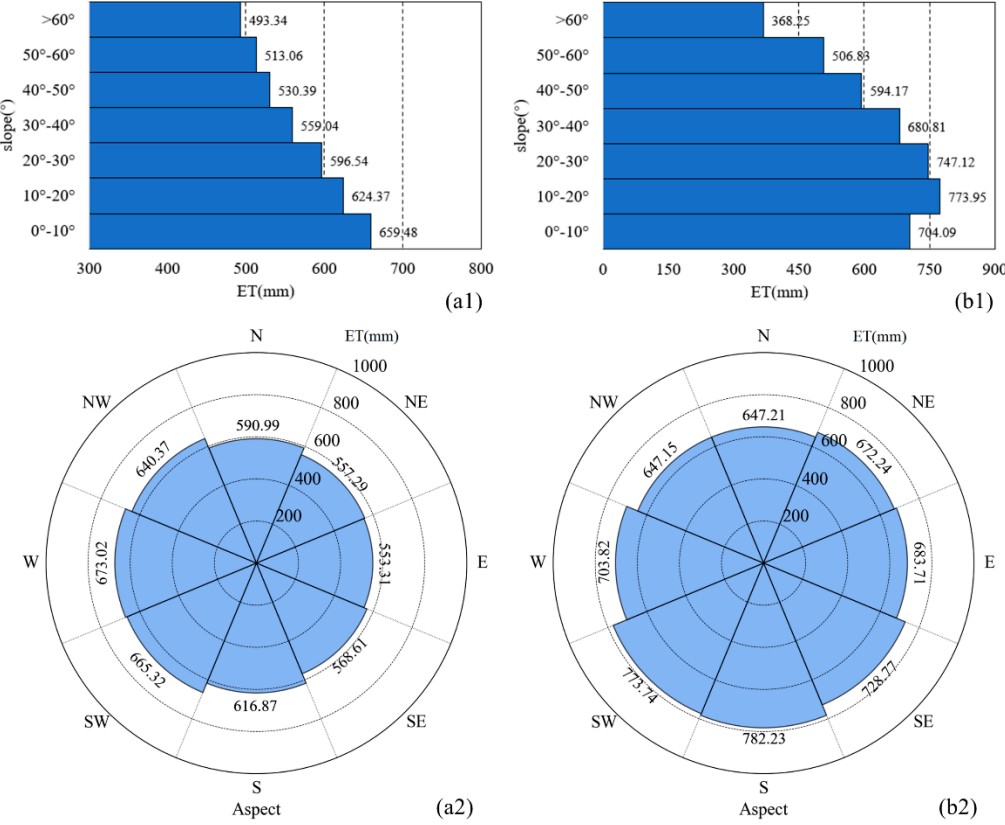

**Figure 20.** Average yearly ET at the different slopes (**a1**,**b1**) and aspect ranges (**a2**,**b2**) in HR (**a1**,**a2**) and BTM (**b1**,**b2**). Aspect range abbreviations such as N, NW and W are explained in Table 5.

**Table 5.** Slope orientation, corresponding aspect range and aspect division.

| Slope Orientation | Aspect Range | Aspect Division |
|---|---|---|
| North (N) | $0° \pm 22.5°$ | Shady slope |
| Northeast (NE) | $45° \pm 22.5°$ | Semishady slope |
| East (E) | $90° \pm 22.5°$ | - |
| Southeast (SE) | $135° \pm 22.5°$ | Semisunny slope |
| South (S) | $180° \pm 22.5°$ | Sunny slope |
| Southwest (SW) | $225° \pm 22.5°$ | Semisunny slope |
| West (W) | $270° \pm 22.5°$ | - |
| Northwest (NW) | $315° \pm 22.5°$ | Semishady slope |

*4.2. Uncertainties and Future Research Directions*

Although the validation results indicated that the model performs reliably, there remain certain shortcomings in the proposed framework that should be further investigated, as suggested below.

For the estimation of direct and diffuse radiation for horizontal surface in this study, we used ground-based observations to empirically regress the fraction of direct radiation and diffuse radiation with relative sunshine duration hours, respectively, and selected the model with the best performances for further estimation. The empirical regression method was chosen for this study because of the sufficient observation data. However, in the actual case, when the radiation from the sun is transmitted to the surface, it is partially absorbed by gases such as water vapor and ozone. Besides, the dust, molecules and cloud droplets in the atmosphere also have a scattering effect on the solar radiation. In future studies, models that take into account the actual transmission should be used to estimate direct solar radiation and sky diffuse radiation [99–101].

Many existing ET models are based on the energy balance residual concept [102–104], such as the ETWatch model. These models require the land surface temperature (LST) as input, which can be limited by the accuracy and spatial resolution of thermal infrared remote sensing data. LST data with a 1-km spatial resolution retrieved from MOD11A1 instead of LST Landsat data were obtained in this study due to the better data availability. Recent studies, such as gap-filling of fine-spatial resolution LST data [105] and spatial sharpening of coarse-resolution LST data [106], have improved the availability and quality of LST data. These key findings should be considered to enhance the prospects of energy balance-based ET models.

In the net radiation calculation model proposed in this study, only the effect of the topography on the downward shortwave radiation was considered. In regard to longwave radiation, the main influencing factors included the cloud cover, surface emissivity, relative humidity, surface temperature, and local atmospheric circulation [107,108]. Considering that the topography also influences longwave radiation to a certain degree, the study of Yan et al. (2020) [109] can be incorporated in future model development to consider the influence of the topography on the directionality of longwave radiation and anisotropy in thermal radiation to obtain terrain-corrected net longwave radiation data, which can then be applied in net radiation calculation.

Soil moisture is an important factor in the process of daily surface conductance reconstruction. The SMAP data considered in this study, with a coarse resolution (10 km), can hardly reflect the soil moisture variation among different underlying surface types and topographic environments. Studies have been reported combining Sentinel-1 and Sentinel-2 images to simulate soil moisture with a high spatial resolution [110–112]. Combining these studies should be considered in future studies to introduce high-spatial resolution soil moisture information into ET calculations.

The NDVI is an important parameter in the ET estimation process. Optical remote sensing data can only provide cloud-free daily observations, and the daily NDVI must be obtained through interpolation. The revisit period is generally long for remote sensing data with a high spatial resolution, but the available cloud-free data are often minimal. As such,

the daily NDVI obtained via interpolation can overlook information on vegetation status changes on cloudy and rainy days, resulting in the situation in which the interpolated NDVI variations differ from the actual changes. Microwave remote sensing data are generally not affected by atmospheric conditions (clouds and aerosols) and can achieve full-time, all-weather monitoring of vegetation conditions. In future studies, the radar vegetation index obtained from Sentinel-1 [113] should be considered in ET calculations instead of the NDVI. We applied Sentinel-2 only to acquire vegetation information, while the potentials of the Harmonized Landsat and Sentinel-2 (HLS) dataset could be explored in future studies. The HLS dataset incorporates surface reflectance data from both Sentinel-2 Multi-Spectral Instrument (MSI) and Landsat-8 Operational Land Imager (OLI) and can achieve land surface monitoring at the spatial resolution of 30 m every 2 to 3 days [114]. More details on vegetation growth conditions could be found based on HLS datasets, which are beneficial to ET studies over complex terrain.

## 5. Conclusions

In this study, a net radiation model considering complex topographic conditions was incorporated into the ETWatch model, and net radiation and ET results with a 10-m spatial resolution were obtained by combining high-spatial resolution remote sensing Sentinel-2 images with meteorological data in the two study areas, i.e., HR and BTM. The ET results were validated against situ observations. The $R^2$ value was 0.84 in HR and 0.86 in BTM, while the RMSE was lower in BTM (0.59 mm) than that in HR (0.82 mm). The model accuracy was satisfactory, better than ET results with a no-terrain-considered Rn model, and the day-to-day variation in the model results also suitably agreed with that in the in-situ observations. The inclusion of topographic factors, such as the slope and slope direction, ensured that the ET results reflected certain topographic characteristics. The mean annual ET value on sunny and semisunny slopes was 616.93 mm, and the mean annual ET value on shady and semishady slopes was 596.22 mm in HR. In contrast, the mean annual ET value on sunny and semisunny slopes reached 761.58 mm, and the mean annual ET value on shady and semishady slopes reached 655.53 mm in BTM. High spatial and temporal resolutions of ET data in mountainous areas with a complex terrain could contribute to a greater understanding of the characteristics of heat and water fluxes at different vegetation growth stages and substrate types and guide ecological protection and rational allocation of water resources in mountainous areas.

**Author Contributions:** L.W. was responsible for the experimental design, manuscript preparation and data processing and presentation. B.W. contributed to the conceptual design, manuscript review, funding acquisition and project administration. A.E., Z.M., S.L., X.N., N.Y. and W.Z. contributed to data processing and manuscript review. All authors have read and agreed to the published version of the manuscript.

**Funding:** This research was financially supported by the National Natural Science Foundation of China (Grant No: 41991232).

**Data Availability Statement:** Not applicable.

**Acknowledgments:** The authors express their appreciation to the data providers in this study. Special thanks go to the Research Institute of Forest Ecology, Environment and Protection and Research Institute of Forest Resources Information Techniques of the Chinese Academy of Forestry for providing the Baotianman station data. The reviewers and editors are also acknowledged. The provided suggestions and efforts greatly enhanced the quality of the paper.

**Conflicts of Interest:** The authors declare that they have no conflict of interest.

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
