# Peer review of "Incorporation of Net Radiation Model Considering Complex Terrain in Evapotranspiration Determination with Sentinel-2 Data"

_remotesensing, doi:10.3390/rs14051191_

Round 1

Reviewer 1 Report

The topic selected by the authors is very interesting and relevant to current challenges in ecohydrology however the manuscript needs to be revised substantially before considering it for publication.

There is a need to improve the manuscript before considering it for publication. In the entire manuscript, there are few points which authors need to consider as it is important in studying the feedback between vegetation and ET especially in these areas which are energy limited ecosystems. Though authors have identified the research gaps the literature survey part can be more streamlined and while coming towards the problem statement. The introduction section need some rework and restructuring to make a precise outline of study.

Currently, some of the statements are not supported by published works. Authors may like to find studies in line with their statements to add scientific weight to their observations. I believe that after duly addressing the comments authors can improve the quality of the manuscript substantially to make it more insightful. I would highly recommend and would be great to mention a line in the assumption to include that “The topographic effect can bring the noise to ET retrieval, especially for these areas. According to a previous study, the topographic effect can be reduced by band ratios, due to the spectrum similarity between NIR and visible bands [https://doi.org/10.3390/cli9070109; https://doi.org/10.3390/rs12020211]”.

I have a big concern in the Introduction, as the authors have missed providing detailed discussion on the important aspect of different classification of ET estimation methods. There is a vast literature on this I would like to suggest few lines following this which author should add is “The ETo estimation models available in the literature may be broadly classified as (1) fully physically-based combination models that account for mass and energy conservation principles; (2) semi-physically based models that deal with either mass or energy conservation; and (3) black-box models based on artificial neural networks, empirical relationships, and fuzzy and genetic algorithms”. I highly recommend adding these recent reference to add more scientific weight in their Introduction (https://doi.org/10.1061/(ASCE)IR.1943-4774.0001199; https://doi.org/10.1016/j.jhydrol.2009.04.029). It is always a good idea to consider the finer resolution satellite than the coarser one that can give a better advantage of capturing the vegetation pattern and thereafter the phenology. However, there is a noise (atmospheric, aerosols, etc.) associated with it so how authors have addressed that problem in their study. 

Also, it would be great if authors can comment that how the data was processed as in the methodology section it is not clear. For instance, Sentinel has many issues related to BRDF or MAIAC corrections so did authors perform any kind of such corrections. If not then please argue that. In addition, the sensitivities of different bands to interfering factors like soil background, atmospheric conditions, BRDF effects, and surface topography need to be discussed.

Authors have missed to include the drawbacks of other satellite products such as AVHRR, MODIS, LANDSAT and Himawari based indirect vegetation proxies. Why do authors not use HLS datasets for these analysis as it has finer resolution and can capture the vegetation phenology very well and nicely.

I encourage and recommend the authors to incorporate a comprehensive detailed review related to the use of satellite-based remote sensing products in the estimation of vegetation growth. Authors may like to find studies in line of their statements to add the scientific weight in their observations. There is a vast literature on this topic.

Please also highlight the correlations result between the Vis and some of the hydro-climatic factors.

Author Response

Dear repsected reviewer,

Please find the response to your comments in the attachment.

Best Regards,

Linjiang

Reviewer 2 Report

Linjiang Wang, Bingfang Wu *, Abdelrazek Elnashar, Weiwei Zhu, Nana Yan, Zonghan Ma, Shirong Liu, Xiaodong Niu . Incorporation of Net Radiation Model Considering Complex Terrain in Evapotranspiration Determination with Sentinel-2 Data

First, I am not an expert in remote sensing. However, I hope I can contribute something to the article in spite of not knowing too much about remote sensing.  I did some modelling of runoff from land, so I was interested in the topic.

Summary. This article appears to address a useful topic: How to improve a model for calculating evapotranspiration on areas with high reliefs slopes (0o to >60o). They find that two measures, the ordinary linear regression (OLR) R2 between ground truth (or ground based observations, line 545) precipitation values and their model calculations are 0.84 to 0.86 for two regions respectively, and the root mean square error (RMSE) were 0.59mm (   ) and 0.82mm ( ) respectively; the default RMSE for a simple model  without “Incorporation of Net Radiation model”(??)  in parentheses. As you see, I had to speculate a little.

Both ground truth and the model showed that the ET was  higher on South sloping hills than on North facing hills, ET calculated for slope brackets 0o-10o and up to >60o

Interpretation of the paper. I have interpreted their paper as an effort to improve upon one standard model by incorporating a net radiation mode (the title), however, if that is the case, I would have anticipated a comparison with the original model results without net radiation. They may be there, but I could not find them.

I very much liked that the study compared two regions, it puts the results into a perspective, and it says something about uncertainties.

The paper is very long, and I understand that it must be long, but as far as I can see, there is not enough information to – in theory- duplicate the results. Equation 4, for example, has coefficients from a to d, but there are no numbers given to the coefficients. I suppose the coefficients would be the result of some calibration, but it is not clear to me how that calibration should be carried out. However, it may be embedded in the text lines 311 to 330 that describes the 10 sub models that the ETWatch model consist of.

There is a Table 1 with numbers for Bands giving some coefficients that relate to Eq. (10). I suppose the information in Table 1 is more important than information on the other coefficients would have been. However, there is a reference to Li et al (ref 52), so I suppose the values could have been retrieved from that publication.

Text. I think the language in the text is quite good, but there are some repetitions that could be removed. (I am not a native -English speaker). If someone else than the authors take on the task to read the article through, I think he/she would be able to remove the repetitions.

Figures. Generally, the figures are quite good and informative. However, they are often tiny, and the text is often difficult to read. I think there are two things that can (easily) be done to improve the readability of the Figures. The letters should be larger, and acronyms used more frequently. For example, in Figure 13 you include the year on the x-axis, but it is the same year for along the x-axis, so the month will do (as in Figure 14).  In figure 13 it is not easy, maybe impossible, to distinguish the lines. The letters on the axes are too small, but then the text is not very sharp, so maybe in the printed document it will be better. Normally, in Figure texts you would have a leading text that apply to all subfigures, and then each subfigure would have started with a letter, a).. b).. . In Figure 6, for example the letters a and b are tiny, but, yes, I could see them.

Figure 3. I use glasses, but I could not read the text. I see the point of presenting the figures as you do, but I suppose other people also would need to use magnifying glasses.

Figure 9 and Figure 13. Better remove years on the x-axis. And I am not able to see the significance between the three curves presented from the Figure.

Figure 10. Are the results here interesting enough to be presented in the abstract?

Figure 18. Magnifying glasses required. But maybe it will be better in the printed version. Could the acronyms along the x-axis be shortened?

Tables. I would suggest that the information on pp. 3 to 5 on the regions could be presented in a table. Then it would be easier to compare the two regions and their characteristics.

Table 2. I think you should use the adjusted R2 for these equations. I am not sure the difference between  R2 for the quadratic and the cubic equations are significantly different. Furthermore, if it is possible, how would you interpret the information you get from R2 and RMSE?  

Minor comments

Line 18 primary methods → primary mechanisms ?

Line 22. “our” model? Is it the ETWatch with or without incorporation- with, I suppose?

Line 28. You use “regarding” very often, may be change a little

Line 69 why is lysimeters outlined?

Line 82 Space

Lines 112-115. Repetitions, could be removed?

Line 205. ... the results are shown in Fig 2, but then you have a reference to literature.

Lines 270-277. I have not gone through all the equations, but I read through Eq(4), and it looks OK to me. Third order term for the regression model for the slope orientation /Ra?  It would have been nice to see what the 3rd order term contributed, but I see that that would b far outside the scope of the article. OK, you comment on it later.

Line 358  As shown in Fig. 6a2, I think it would be better with just a, b, c..

Line 372, the Fig 7. The red, green, and red..→ The red, green, and ?? The blue regression looks adequate.

Line 387. It is very nice that you show the scatterplots. Text on x-and y- axes could be larger.

Line 590. I tried to make a Table that presented the results very clearly. I think it is possible. From the title, I would think that “Incorporating of Net..” is an important issue. Maybe it should be addressed more explicitly?

Author Response

Dear respected reviewer,

Please find the response to your comments in the attachment.

Best Regards,

Linjiang

Round 2

Reviewer 1 Report

I recommend the manuscript to be published as authors have addressed all the comments.

Reviewer 2 Report

I found the response to all my questions and comments adequate. And I agree with you on the one advice you did not follow.